# Measurement Technologies of Light Field Camera: An Overview

**DOI:** 10.3390/s23156812

**Published:** 2023-07-31

**Authors:** Xiaoming Hu, Zhuotong Li, Li Miao, Fengzhou Fang, Zhongjie Jiang, Xiaodong Zhang

**Affiliations:** 1State Key Laboratory of Precision Measuring Technology & Instruments, Laboratory of MicroNano Manufacturing Technology, Tianjin University, Tianjin 300072, China; huxiaoming@jumperscience.com (X.H.); lzt_1721@tju.edu.cn (Z.L.); miaoliii@126.com (L.M.); fzfang@tju.edu.cn (F.F.); 2Beijing Jumper Science Ltd., Beijing 100036, China; jiangzhongjie@jumperscience.com

**Keywords:** light field camera, calibration technology, reconstruction algorithms, measurement application

## Abstract

Visual measurement methods are extensively used in various fields, such as aerospace, biomedicine, agricultural production, and social life, owing to their advantages of high speed, high accuracy, and non-contact. However, traditional camera-based measurement systems, relying on the pinhole imaging model, face challenges in achieving three-dimensional measurements using a single camera by one shot. Moreover, traditional visual systems struggle to meet the requirements of high precision, efficiency, and compact size simultaneously. With the development of light field theory, the light field camera has garnered significant attention as a novel measurement method. Due to its special structure, the light field camera enables high-precision three-dimensional measurements with a single camera through only one shot. This paper presents a comprehensive overview of light field camera measurement technologies, including the imaging principles, calibration methods, reconstruction algorithms, and measurement applications. Additionally, we explored future research directions and the potential application prospects of the light field camera.

## 1. Introduction

Machine vision measurement is an effective industrial inspection technology that offers the benefits of fast speed, high accuracy, and low cost. It also has great development potential and a wide range of prospective applications. Machine vision measurement is currently employed extensively in the fields of aerospace, biomedicine, agricultural production, and social life [1,2,3,4]. For instance, in the aerospace industry, the positional estimate of cooperative targets is accomplished using visual measuring systems [5]. Light detection and ranging (LiDAR) is used to acquire the position of non-cooperative space targets with an accurate result [6]. However, the applications of each of the aforementioned measurement techniques have some drawbacks. Table 1 lists the advantages and disadvantages of them. Single camera systems, despite their compact size and light weight, have a limited ability to measure depth and must be supplemented by displacement motion mechanisms, which will increase the complexity of the measuring system and the measurement time [7]. Multi-view systems can measure 3D objects with a high degree of precision through multi-camera calibration and matching algorithms. However, each time the system is set up, the multi-view system must be re-calibrated and the relative position must be stable. Additionally, multi-view systems are expensive and bulky [8]. LiDAR systems offer great measurement precision, but they are expensive and heavy and their sampling rate low [8]. The requirements for measuring efficiency and measurement system size are challenging for such measurement systems to achieve. With continuous research on light field theory and light field acquisition methods, the concept of the light field camera has been proposed and applied to the measurement [9]. A light field camera has the benefit of being compact and reconstructing the 3D shape of the surface with just one shot. At the same time, it can provide dense multiple angles and obtain richer image information, improving measurement accuracy and solving measurement challenges such as occlusion. However, the light field cameras suffer from major measuring shortcomings as a result of their unusual design. The first is the trade-off that light field cameras make in terms of spatial resolution in order to be able to estimate depth, which in theory lowers the measurement accuracy and becomes the main issue impeding the development and application of the light field camera. At the same time, light field images often have very high resolution and require the use of complex image-processing techniques, which lead to problems such as memory usage and longer computational times. Therefore, one of the main research questions in the field of light field camera measurement is how to increase the measurement accuracy, as well as the algorithmic efficiency.

The accurate calibration of the camera and the accurate 3D recovery from the light field image are the foundation for accurate measurements using light field cameras. To increase the precision of the ensuing solution, it is necessary to calibrate the essential parameters against the theoretical differences after the light field camera has been built. At the same time, the accuracy and speed of the measurement are significantly influenced by the accuracy and effectiveness of the reconstruction algorithms. As shown in Figure 1, this paper first introduces the structure and measurement principles of the light field camera, then conducts a thorough investigation into the calibration methods of the light field camera and the reconstruction algorithms of the light field camera and summarizes the measurement examples of the light field camera according to its characteristics.

## 2. Imaging Principle of the Light Field Camera

### 2.1. The Structure of the Light Field Camera

The ability of light field cameras to achieve three-dimensional measurements from a single image is inextricably linked to their special structure. The structural changes to the light field camera are shown in Figure 2. Adelson et al. [14] was the first to conceptualize the light field camera, placing a tiny pinhole array on the image plane to analyze the structure of light at each of its macroscopic pixels. However, the small light flux of the pinhole arrays did not allow for good imaging. With the advancement of microlens array fabrication technology, Ng et al. [9] created the first standard light field camera, by swapping the prior pinhole array for a microlens array. Since then, researchers have begun to pay more and more attention to light field cameras and have started to use them for measurements.

The disadvantage of the standard light field camera is that its matching spatial resolution is decided after the size of the microlens array is established. The size of the microlenses must be made extremely small in order to achieve a higher spatial resolution, which, on the one hand, will significantly increase the processing requirements for the microlens array and, on the other, because of the diffraction limit and the edge effect of the microlenses, will degrade the imaging quality. To solve this problem and improve the spatial resolution of the light field camera, Georgiev et al. [15] proposed the focused light field camera. It moves the microlens array from the image plane to a location where the image from the main lens can be secondarily imaged to the image sensor. Based on where the picture plane of the main lens is located, the focused light field camera can be classified as a Galilean or Keplerian structure [16], both of which have essentially the same operating principles and solution process. The focused light field camera’s spatial resolution no longer just depends on the size and number of microlenses, but also on the relative placements of the main lens, microlens array, and sensors. The main problem with focused light field cameras is the small depth of field, for which Georgiev et al. [17] proposed the concept of a multi-focal light field camera, which uses staggered microlens arrays of different focal lengths, focused on two or more different planes on top of the focused light field camera. In this way, a focused image can be constructed at a greater depth of focus, enhancing the depth of field of the system. However, accordingly, the system resolution decreases.

The essence of the light field camera is to use a microlens array to compromise the spatial resolution for the angular resolution and obtain the capability to collect images from multiple angles. The three types of light field cameras are a trade-off in terms of the relationship between some parameters. For the same image sensor and main lens, a greater angular resolution often means a higher depth resolution and a greater spatial resolution often means a higher lateral resolution [10]. S. Zhu et al. [18] compared the standard light field camera and the focused light field camera through experimental and modeling analyses, as shown in Table 2. The focused light field camera has a higher spatial resolution and reconstruction accuracy than the standard light field camera, but a smaller depth of field and a lower angle resolution. Multi-focal light field cameras are very similar in performance and principle to focused light field cameras, with the difference that their use of microlenses with different focal lengths improves the depth of field of the measurement, but relatively reduces the spatial resolution. As a result, we may build and select the appropriate type of light field camera based on the actual measurement requirements and processing conditions.

### 2.2. Measurement Fundamentals

The light field camera can realize the segmentation of the main lens by increasing the microlens array so that it can record the result of observing the object from multiple angles, and then, it can realize the function of three-dimensional measurement. Different types of light field cameras vary in the segmentation of the main lens due to the different positions of their microlens arrays, and their basic principles are as follows. (The principle of the multi-focal light field camera is similar to the focused light field camera, except that different focal lengths of the microlens arrays are chosen for different depths of the object for the calculation.)

The basic principle of the standard light field camera is shown in Figure 3a. The microlens array is placed at the image plane of the object point, and the light from Point A converges on the microlens array, which is further differentiated according to the direction of the light. To maximize the angular resolution, we want the sharpest microlens images possible, so the sensor is fixed at the focal length f_*m*_ of the microlens array. The standard light field camera divides the main lens by different pixels behind the same microlens, and the pixels at the same position behind each microlens are the projections of the same sub-aperture of the main lens. These pixels are sequentially arranged to collectively form a sub-aperture image. Therefore, the angular resolution of a standard light field camera depends on the number of pixels corresponding to the microlens, and the spatial resolution is the number of microlenses.

The basic principle of the focused light field camera is shown in Figure 3b, where an array of microlenses is placed behind the image plane of the object point, enabling secondary imaging of the image from the primary lens to the sensor position. The light from Point A converges before the microlens array and continues to propagate forward to be imaged by different microlenses. The point is imaged secondarily by different microlenses to split the primary lens. Thus, its angular resolution depends on how many microlenses the image point can be secondarily imaged by, and no longer on the number of pixels corresponding to a single microlens. The angular resolution in one direction is a/b, and the distance of the primary image plane from the microlens array is a, while the distance of the sensor from the microlens array is b. They satisfy the Gaussian imaging relation for the focal length of the microlens.

Furthermore, the light field camera can be considered a camera array and is able to recover images from different views. In this section, we provide a concise overview of the light field camera method for recovering multi-view images, along with the computation of the baseline and focal length for a comparable camera array. By converting to a camera array, the readers will gain a more-intuitive understanding and simplified calculations of the measurement system’s parameters, including the depth of field, resolution, field of view, and so on.

Figure 4a shows the corresponding camera array calculation for a standard light field camera. Each pixel behind a microlens corresponds to a view, and the combination of pixels corresponds to the same position of each microlens in an image in one viewpoint. As a result, the number of cameras is equal to the number of pixels corresponding to one microlens imaging region. Therefore, the equivalent camera’s optical center is defined as the intersection of the imaging rays of a sub-aperture image at the main surface of the main lens. Furthermore, the geometrical relation can be used to compute the baseline length b_*L*_ and the focal length f of the camera array, which can be found in the literature [19,20].

The equivalent camera array calculation of the focused light field camera is shown in Figure 4b. The different microlenses of the focused light field camera observe the point of the primary imaging plane, which can be directly equated to the camera. Subsequent transformations can further equate the light field camera to a camera array. This figure only traces the light to the primary imaging plane of the main lens. The primary imaging plane is continuously divided according to the size of the microlens array, and each region imaged by the microlens array will be a patch. Different division (red, green, and blue) corresponds to a different patch. By combining the patches of the same segmentation in order, we can obtain an image from one viewpoint. The camera baseline b_*M*_ and the focal length f_*m*_ in this figure are equal to the diameter of the microlens (d) and the distance b between the microlens and the sensor, respectively. The number of equivalent cameras in one direction is a/b. The equivalent camera array parameters of the whole light field camera can be subsequently obtained by geometric transformation.

### 2.3. Light Field Camera Design

To ensure the measurement accuracy and the proper application of the light field camera, it must be designed appropriately. The selection of the parameters of the light field camera was well described by M. Diebold et al. [21] and S. Shi et al. [22]. We introduce it briefly. The structural design starts with an assessment of the measurement needs, weighing the trade-offs between spatial and angular resolution, then selecting the appropriate type of light field camera. After that, the core components of the light field camera such as the image sensor, the microlens array, and the main lens are selected and designed. The selection of the main lens is often based on the working distance, and it is preferable that the aperture be adjustable to satisfy the F-matching between the main lens and the microlens array [9]. Diebold equated the light field camera to a camera array and limited the parameters of the light field camera from two perspectives, which are the depth and the field of view. The first is the constraint of depth versus parallax. For instance, to compute orientation using the structure tensor, the disparity must lie in a symmetric 2 px range. This parameter restricts the camera array’s focal length, baseline, and pixel spacing. When they are fixed, the object must be in the range from Z_+_ (distal) to Z_−_ (proximal). In addition to the depth constraint, a specific field of view within the measurement distance is also considered to ensure that all cameras can observe all parts of the target scene. Furthermore, as shown in Figure 5, a Frustum space can be defined to determine the image space of the light field camera so that it contains the object to be measured. Not only that, after meeting the above conditions, it is also necessary to consider the sensor frame rate, as well as the resolution requirements. After determining the sensor size to meet the field of view, the sensor resolution should be selected as large as possible to enhance the measurement accuracy. However, it should be noted that, often, the frame rate of high-resolution cameras will decline. Therefore, it is not very practical in some of the requirements of high dynamic measurement occasions. Therefore, the trade-off between the frame rate and camera resolution also needs to be made according to the measurement needs. In addition, Shi et al. also discussed the mechanical structure of the light field camera mounting design, which can be referred to by readers who need it.

## 3. Calibration Technology of the Light Field Camera

The light field camera comprises a main lens, a microlens array, and an image sensor. In the actual measurement process, obtaining accurate 3D information is intricately tied to the size and position of the microlens array, as well as the focal length and aberration of the main lens. For instance, the distance between the microlens array and the sensor plane directly impacts the disparities at the same point. Additionally, the focal length of the main lens significantly affects the measurable range and lateral resolution. Furthermore, Thomason et al. [23] demonstrated that the deviation of the microlens array from its theoretical position greatly influences the reconstruction error. However, in practical machining and assembly processes, these crucial parameters may deviate from their theoretical design values, thus impacting the measurement accuracy. Hence, it becomes necessary to calibrate the light field camera to obtain the precise values for each parameter before it can be utilized for actual measurements. Calibration methods for the light field camera have been extensively researched and can be categorized into three main types: calibration based on the light model, calibration based on the raw light field image, and calibration directly on the depth calculation model. The following will provide a description of each of these three methods.

Calibration based on a light model involves tracing light from the object space, passing through the main lens, the microlens array, and the image sensor in sequence. The light model of the light field camera is shown in Figure 6. Dansereau et al. [24] combined the structure of a light field camera with the influence of lens aberrations. In this approach, the main lens is represented using a thin lens model, while the microlens array is modeled as a pinhole. A novel 15-parameter light field camera intrinsic reference model (10 intrinsics + 5 distortion) is proposed, employing a four-dimensional internal reference matrix to establish the relationship between each recorded pixel and its corresponding light rays in 3D space. The calibration of the light field camera is then conducted based on this model. This research represents a significant milestone in the modeling and calibration of light field cameras, as it offers a more-detailed and precise description of the intrinsic parameters compared to previous models. However, it should be noted that the model does not account for the effects of depth distortion, microlens tilt, and so on. Furthermore, it is only designed and calibrated for standard light field cameras. Building on its foundation, Johannsen et al. [25] introduced a new model correcting for radial lateral, as well as radial depth distortion and successfully constructed and calibrated the focused light field camera. Subsequently, Thomason et al. [23] considered misalignments of the microlens array and provided reasonable estimates for the position and orientation of the microlens array, which could be applied to any light field camera. Expanding the calibration method further, Heinze et al. [26] extended it to the multi-focus light field camera and introduced a model for main lens tilt and offset in the subsequent study [27]. To enhance the calibration model of the light field camera, Zhang et al. [28] introduced a more-detailed geometric model of the microlens array. This model was used to calibrate the non-planarity of the array and address misalignment issues, resulting in more-accurate decoding of the light field for focused light field cameras. Subsequent researchers have continued to optimize the calibration model. For instance, Zhou et al. [29] proposed a model that represents the light field camera using parameters with specific physical significance. Duan et al. [30] drove a homogenous intrinsic matrix, which represents the relationship between each pixel on the image plane and its corresponding light field. However, it is worth noting that all current models based on light field cameras are constructed with a pinhole model instead of a microlens array and a thin lens model instead of the main lens. This substitution can lead to increased calibration errors, particularly at the lens’s edges [23,28]. Consequently, future studies should focus on optimizing these areas to further reduce calibration errors.

The second type of calibration is based on the raw light field image. Most calibration methods for light field cameras are conducted using sub-aperture images, which requires a good estimation of the intrinsic camera parameters. However, this approach often encounters the challenge of recovering images with unknown parameters. To overcome this issue, Bok et al. [31] introduced a method to directly calibrate light field cameras using the raw light field image. One main difficulty with this approach lies in the direct extraction of target features. To tackle this challenge, researchers have extensively explored feature extraction methods directly from the original light field image. The typical feature extraction is shown in Figure 7, including points, lines, as well as discs. Bok successfully extracted line features from the original light field image for light field camera calibration. Furthermore, Noury et al. [32] and Liu et al. [33] utilized a sub-pixel corner-point-detection method to extract corner point features from the original image. Additionally, Brien et al. [34] proposed a new feature type for light field cameras, named the light field disc feature, and presented a corresponding model utilizing disc features for 3D spatial point transformation. This method has demonstrated successful application in various types of light field cameras.

The third type of calibration involves directly calibrating model parameters such as multiple cameras or virtual depths to obtain the necessary parameters for the subsequent light field reconstruction. This approach establishes a direct relationship between the depth solution model and the calibration parameters. Extensive research has been conducted in this area, resulting in the development of various calibration models such as two-plane calibration models, imaginary depth calibration models, sub-aperture calibration models, and polar plane calibration models. Figure 8 illustrates some typical light field camera models. Zhang et al. [35] proposed a multi-projection center (MPC) model based on two-parallel-plane (TPP) representation, which described different types of light field cameras using a generic six-intrinsic-parameter model. Zeller et al. [36,37,38] introduced a calibration model specifically for a focused light field camera, which established a relational calibration between distance and virtual depth. By recovering the sub-aperture image at different sub-apertures of the light field image, depth recovery can be achieved through visual matching. Expanding on this concept, Aggoun et al. [39] estimated and calibrated the position and baseline of the virtual lens in the corresponding camera array. Monteiro et al. [40] further refined the calibration model, addressing limitations such as the complete viewpoint projection matrix for the definition and consistent lens parameters. In the case of the epipolar-plane-image (EPI)-based recovery method, Zhou et al. [41] proposed a geometric model known as the epipolar space (EPS), which directly determines the relationship between the 3D points to be measured and the corresponding parameter vectors in the polar space through calibration.

The study of light field cameras extends beyond the aforementioned results, with numerous studies and innovations focusing on parameter separation, adaptive calibration without calibration plates, and projection assistance, among others. Strobl et al. [42] introduced a decoupling method for calibrating the traditional capabilities of light field cameras from the calibration of their novel features. This approach ensures that the higher noise levels associated with the novel features do not affect the estimation of traditional parameters such as the focal length and radial lens distortion. Ji et al. [43] proposed a calibration method for light field cameras that eliminates the need for a calibration target, achieving self-calibration. Cai et al. [44] proposed a calibration method for structured light field systems. The calibration of a structured light field system is accomplished through structured light assistance. To enhance the accuracy of microlens array center detection in light field cameras, Chen et al. [45] devised a novel fuzzy circle detection algorithm, improving the precision of extracting the circle center from the light field white image. Wei Feng et al. [46] presented a calibration method based on edge diffusion, which utilizes the effect of the extracted pixel block size of the light field camera on the continuity of the sub-aperture image, thus achieving camera calibration.

On the one hand, calibration methods for light field cameras are advancing in sophistication, with an increasing number of calibration parameters. Parameters such as the main lens focal length, aberration, orientation, and microlens array orientation are being optimized, providing a solid foundation for achieving high-accuracy measurements with light field cameras. In the future, it is expected that the limitations of small-aperture and thin-lens models can be overcome, bringing light field camera calibration closer to real-world applications. On the other hand, the direct extraction of features from the original image for calibration, without relying on primary parameters, has its challenges. The complexity of feature extraction can lead to a degradation in calibration accuracy [47]. Therefore, for calibration without primary parameters, a calibration method based on the original light field image can be employed to obtain more-desirable primary parameters. Subsequently, the sub-aperture image can be used for secondary calibration to further improve accuracy. Furthermore, as the application domain of light field cameras expands, calibration methods tailored to different measurement situations will be researched and proposed, taking into account various application scenarios and measurement requirements. For example, J. Sun et al. [48] and S. Shi et al. [49] proposed light field camera calibration methods applied to flame temperature measurement and particle image velocimetry, respectively. Overall, the continuous advancement of calibration techniques (addressing limitations and adapting to specific needs) will contribute to the wider adoption of light field cameras in diverse fields of application.

## 4. Reconstruction Algorithms of the Light Field Camera

The recovery of 3D depth from a single light field image is a great advantage of light field cameras and the basis of measurements, and reconstruction accuracy is also directly related to the measurement accuracy of light field cameras. On the one hand, due to the special structure of the light field camera, the extension of disparity space into continuous space can improve the robustness and accuracy of depth estimation [50]. On the other hand, the decrease in accuracy due to the short sub-aperture baseline of the light field camera and the increase in computational time due to the processing of high-dimensional light field data are the main challenges faced by light field camera reconstruction. Therefore, fast and highly accurate reconstruction algorithms that combine the advantages of dense sampling of the light field camera and compensate for their shortcomings are the main focus of the research [51].

### 4.1. Light Field Reconstruction Based on Traditional Algorithms

After the introduction of the light field camera concept, Georgiev et al. [15] initially employed correlation techniques to find matching points between apertures for preliminary depth estimation. However, the reconstruction accuracy of this approach was limited. To enhance the accuracy of light field image reconstruction, stereo vision methods were applied. Given the dense view provided by the light field camera, reconstruction constraints should consider all available views. Yu et al. [52] pioneered the use of stereo matching to estimate parallax and calculated depth by leveraging the geometric triangular structure of 3D rays in space. Heber et al. [53] performed image warping from multiple views to a central view and achieved depth recovery using principal component analysis. Bishop [54,55], Sabater [56,57], and others further optimized reconstruction methods by addressing anti-aliasing filtering and demosaicing issues. However, the aforementioned methods stay at the pixel level of disparities and are limited by the baseline length of the light field camera, so their reconstruction accuracy is not high. To tackle this problem, Jeon et al. [58] introduced the phase shift theorem in the Fourier domain to estimate sub-pixel shifts in sub-aperture images. Additionally, gradient cost aggregation, utilizing the angular coordinate adaptation of the light field, was employed for sub-pixel-level parallax estimation through optimization. These advancements aimed to improve the accuracy of light field image reconstruction by addressing limitations and introducing more-refined techniques at the sub-pixel level.

In addition to providing dense multiple views, the isometric rectangular array arrangement of the equivalent camera array in the light field camera offers new cues and constraints for depth recovery, thus ensuring reconstruction accuracy, as shown in Figure 9. Tao et al. [59] combined defocus and corresponding depth clues to estimate depth. Subsequent studies introduced shadow depth estimation cues [60] to further enhance the accuracy of depth estimation. Lin et al. [61] utilized focal stack rendering as a data consistency metric, leveraging the symmetry of non-occluded pixels along the center of the focus slice. This approach remains highly effective even in the presence of noise in the light field sampling. However, it is less robust in regions where occlusion occurs.

The occlusion problem is a significant challenge in visual reconstruction and is inherent in reconstruction. In complex real-world environments, mutual occlusion between objects and self-occlusion of objects can disrupt spatial and angular consistency, thereby impacting reconstruction accuracy and introducing new algorithmic challenges. C. Chen et al. [62] were the first to propose an occlusion-aware algorithm that computed a bilateral metric at every possible depth, finding a local minimum at the true depth. This approach effectively handles severe occlusions. Building upon this work, T. Wang et al. [63] developed a single-occlusion model and proposed an occlusion-depth-estimation algorithm based on improved angular consistency. However, this method is limited to a single occluder and relies heavily on edge-detection results. To address multiple occlusions, H. Zhu et al. [64] derived occlusion consistency between spatial and corner spaces and constructed a multiple occlusion model, as shown in Figure 10. However, they did not evaluate the impact of noise on reconstruction accuracy. In order to achieve robust reconstruction in the presence of both occlusion and noise, Williem et al. [65,66] introduced two novel data costs that combine the robustness to occlusion and noise immunity of the angular image and focal stack image. This approach significantly enhances the algorithm’s resilience to noise and occlusion, leading to improved reconstruction accuracy.

In addition to stereo vision, the arrangement of virtual viewpoints in light field cameras in an isometric linear rectangular array allows for the application of the epipolar plane, bypassing the cost volume construction and consistency constraints. The concept of the epipolar plane for 3D scene reconstruction was initially introduced by Bolles et al. [67], enabling the creation of dense 3D descriptions from image sequences. As shown in Figure 11a, the principle involves taking line samples along the linear arrangement of isometrically arranged cameras, and the object’s depth can be determined based on the slope of the line formed by the object sample. Light field cameras align well with this requirement, and depth recovery algorithms based on polar plane images have been employed in light field image reconstruction as light field imaging technology has progressed. For example, Wanner et al. [50,68] utilized the structure tensor to analyze the orientation of straight lines in epipolar plane images. However, this method only utilizes one-dimensional angular samples and is not effective in the presence of occlusion. Building upon this research, occlusion handling using the polar plane method has also been extensively investigated. Zhang et al. [69] introduced the spinning parallelogram operator (SPO) into the depth estimation framework to mitigate the effects of occlusion and noise. The diagram is shown in Figure 11c, and the slope of the corresponding point can be better extracted. Sheng et al. [70] proposed a strategy for extracting epipolar surface images in all available directions, maximizing the utilization of regular grid light field images, as shown in Figure 11b. This approach overcomes the limitation of previous methods that focused solely on horizontal and vertical EPI, neglecting other directions. By considering the direction along the occlusion boundary for calculation, the accuracy in the vicinity of occlusion is further improved.

The dense multiple views of the light field camera also aid in the reconstruction of non-Lambertian surfaces. As diffuse and specular reflections exhibit different behaviors at different viewpoints, light field cameras possess the ability to recover highly reflective object surfaces [71]. Furthermore, the properties of the light field camera have been explored for measurements in scattering media such as water and fog. Tao et al. [72] developed an iterative method to estimate and remove specular components, improving the accuracy of reconstruction. Then, the relationship between light field data and the reflectivity of a two-color model was utilized to estimate the light source color and separate highly reflective surfaces [73]. Additionally, Wang et al. [74] derived a theory of bidirectional reflectance distribution function (BRDF) invariance for recovering 3D and reflectivity estimations from light field cameras. It partially solved the problem of the shape measurement of glossy objects such as metals, plastics, or ceramics. The effectiveness of the reconstruction is shown in the first row of Figure 12. In the presence of scattering media, Tian et al. [75] proposed a novel non-uniform correction method that surpasses existing techniques in removing back-scatter and attenuation. It is more effective to remove back-scatter and attenuation from images compared with existing methods. The reconstructed results are displayed in the second row of Figure 12.

### 4.2. Light Field Reconstruction Based on Deep Learning

It has been demonstrated [76,77] that the structural-tensor-based algorithm suggested by Wanner [50,68] is the fastest of the traditional algorithms. This is most likely owed to the fact that it only considers one-dimensional angular sampling and does not account for occlusion processing; therefore, its reconstruction is rather fast. Nonetheless, this approach still requires more than 8s to reconstruct, and the reconstruction effect is poor. As the algorithm’s accuracy improves, so does its time consumption, and if comprehensive occlusion modeling is performed, the reconstruction time will take several thousand seconds, making it difficult to meet the needs of real measurement applications. With the continuous development of deep learning techniques, convolutional neural networks (CNNs) have lately been effectively used in many computer vision (CV) applications, and tremendous progress has been made in terms of algorithm efficiency and accuracy improvement. Building on these findings, researchers have applied deep learning networks to light field camera depth recovery, lowering reconstruction time while maintaining reconstruction accuracy.

Heber et al. [78] developed a CNN-based approach using the polar plane method. The network predicted the 2D hyperplane orientation in the light field domain based on the input image, enabling scene geometry reconstruction. However, this method only considers one directional epipolar geometry of light field images in designing the network. Building upon this work, C. Shin et al. [77] addressed the limitation by introducing a multi-stream network that encodes each epipolar image separately to enhance depth prediction. Their method achieved reconstruction in just 2 s. Tsai et al. [79] proposed an attention-based view-selection network that efficiently utilizes information from the epipolar map through an attention map. This method considers more directions than Shin, leading to improved reconstruction accuracy. The reconstruction time for this method was approximately 5.5 s. Apart from deep learning methods based on the epipolar plane, Huang et al. [80] estimated depth maps from focus stacks and proposed a non-iterative low-frequency reconstruction method using three successive CNNs. Their network improved reconstruction accuracy and efficiency, with a reconstruction time of 4.14 s. All of the aforementioned methods employed a supervised training approach, which relies on ground truth disparity for training. However, obtaining accurate depth labels for real scenes is challenging and limits the availability of training data. To address this issue, J. Peng et al. [81] presented an unsupervised depth-estimation method using a CNN, eliminating the need for accurate depth labels during training. This provided a practical solution for applications where obtaining accurate depth information is difficult. However, this method did not consider the effects of occlusion and texture-free regions, resulting in lower reconstruction accuracy. To solve these, B. Liu et al. [82] designed a cascaded-cost-volume network that leveraged geometric features between sub-aperture images to generate accurate parallax maps. The architecture of the network is shown in Figure 13. It was trained using a combined unsupervised loss, incorporating occlusion-aware luminosity loss and edge-aware smoothness loss to improve performance in occluded and texture-free regions, respectively.

### 4.3. Conclusions

B. Liu [82] compared several typical reconstruction algorithms including traditional algorithms by Wang [63] and Williem [66] with supervised-based deep learning algorithms by Shin [77] and Tsai [79] and unsupervised-based learning algorithms by Peng [81] and Liu [82]. The result is shown in Table 3. The parameters for their comparison were Badpix0.07, as well as the MSE. Badpix0.07 represents the percentage of depth values below a relative error based on the ground truth (7%). The MSE represents the mean-squared error between the predicted disparity maps and the ground truth.

As research progresses, there have been significant improvements in the accuracy of traditional light field image reconstruction. These advancements have led to improved robustness against occlusion, noise, and other factors. For instance, Williem’s work [66] achieved an average mean-squared error (MSE) value of 0.85 or less on a dataset with various occlusions. However, traditional methods often involve a trade-off between computational time and reconstruction accuracy, making it challenging to achieve high-accuracy reconstruction while controlling the processing time. Deep-learning-based methods have shown promise in improving reconstruction speed, especially when utilizing GPUs for acceleration. For instance, Huang’s technique [80] achieved an average reconstruction time of 0.062 s with GPU acceleration. However, the reconstruction accuracy of deep learning methods, particularly on real-world datasets, still needs improvement. Currently, unsupervised approaches demonstrate higher stability and accuracy compared to supervised approaches, making them an important direction for future research. CNN-based approaches also face challenges related to large dataset requirements. The currently available publicly accessible light field datasets do not provide sufficient data to train deep networks effectively [77]. Therefore, further improvements and expansion of light field datasets are needed to support the training of deep networks in the field of light field reconstruction. According to the above, the performance comparison between traditional algorithms and deep learning algorithms is shown in Table 4.

In summary, the light field camera reconstruction algorithm has achieved a huge improvement in reconstruction accuracy, which to a certain extent makes up for the problem of accuracy caused by the short baseline of the light field camera. At the same time, with the continuous research of deep learning algorithms, the time required for its depth recovery has been greatly reduced, which has fully secured the application of light field cameras in the field of measurement. However, the traditional algorithms with higher accuracy are currently less efficient, while the deep learning methods with higher efficiency have relatively lower accuracy. Therefore, it is important to perform research to further improve the efficiency of the algorithms in terms of hardware and software with guaranteed accuracy. On the other hand, light field reconstruction accuracy relies on an increase in the number of viewpoints, which will lead to a sacrifice in spatial resolution. Therefore, with changes in structures, achieving high-accuracy depth recovery with the smallest possible number of viewpoints (e.g., 2 × 2) is also an important research element for the future. Finally, as the application fields of light field cameras increase, reconstruction algorithms adapted to different measurement objects will also be studied and proposed one after another. For example, the light field flame temperature field reconstruction method based on optical tomography [83] and the light field particle image velocimetry (PIV) tomography reconstruction technique based on the sense-ray-tracing-reconstruction technique [84], etc., are combined with the actual measurement objects, and new reconstruction algorithms are proposed.

## 5. Measurement Application of Light Field Camera

The light field camera enables high-precision three-dimensional measurements with a single camera through only one shot, without the need for hardware assistance such as displacement tables. Due to these characteristics, light field cameras are used in a wide range of applications where cost and quality control, system volume control, measurement in special environments, dynamic measurement, and identification and classification are required. Currently, Raytrix [11] and VOMMA [85] have developed mature light-field-camera-measurement techniques and products based on focused light field cameras and standard light field cameras, respectively. Table 5 lists the official application product with the maximum field of view or minimum lateral resolution, representing the current leading measurement capability of light field cameras. Not that many researchers have studied it extensively.

A major advantage of the light field camera is that the system is small, lightweight, and easy to build, requiring only one camera for the measurement, without the need for complex multiocular coupling and calibration. Some typical applications of the light field camera based on cost and volume control are shown in Figure 14. The German Aerospace Center [86] applied a focused light field camera to 3D geometry measurement for non-cooperative spacecraft. This is of great importance in the space field as it can effectively reduce the mass of the measurement system compared to traditional LiDAR and multi-view systems. They achieved a lateral and axial accuracy of ±3 mm and ±12 mm, respectively, at a distance of 20 cm from the camera. In order to improve the angular resolution, Xu et al. [10] built a standard light field camera measurement system to improve the z-direction accuracy of the measurement. However, it took about 3 s to obtain the point cloud after using GPU acceleration, which has not yet met the control requirements. In the future, the solution speed can be improved by downsampling or using deep learning algorithms. In addition, the simple construction facilitates the use of light field cameras in some structures with volume requirements. In bubble measurement, a multi-view system would be superior to the traditional combined measurement system of a single camera with a prism or reflector, but cannot be used where optical space is limited. To solve this problem, Chen et al. [87] implemented bubble measurements using a light field camera, achieving higher measurement accuracy than systems using prisms. To improve the accuracy of the lateral positioning of the bubble size and center of mass position, measurements were completed using a focused light field camera. However, the consequence of this is that the measurement accuracy is now highly variable at different depths. Not only that, compared to multi-view systems, light field cameras have better stability, making it easy to architect a stable system and maintain high measurement accuracy in the field or other complex working environments. Schima et al. [12] applied a light field camera to the external structure of plants and changes in plant morphology to improve the stability of the measurement system in the field. Sun et al. [88]. also applied the light field camera to flame temperature field measurement, and the relative error of the method for flame temperature was no more than 0.5%. Y. Luan et al. [89] proposed an optical field multispectral radiation thermometry method based on a standard light field camera. S. Shi et al. [90] used optical field cameras to achieve pressure measurements on complex geometries in fully aerodynamic models without repeatedly installing, aligning, and calibrating multiple cameras in large wind tunnel experiments. The fitting error was less than 1 mm over a circular cross-sectional range of 20–50 mm in diameter, but the current depth measurement accuracy was lower than that of the photographic model deformation measurement method.

Another feature of the light field camera is that it can perform 3D reconstruction with a single photo, making it ideal for dynamic measurements. Some typical applications of the light field camera based on dynamic capabilities are shown in Figure 15. Light field microscopy [91] is one such measuring application in which the microlens array structure is merged with the microscope structure, allowing a 3D reconstruction of the micro-structure to be performed with a single photograph. When compared to light-sheet microscopy, there is no need to scan the sample layer by layer, avoiding the requirement for additional calibration labor and data collecting while displaying higher dynamic measurement capacity. Its accuracy can reach the micron level. Considering its dynamic measurement capabilities, Chen et al. [92] used a light field camera for human iris measurements to enhance the measurement of the iris as a dynamic tissue compared to the commonly used methods of 2D cross-sectional optical coherence tomography (OCT) or ultrasound images. A depth resolution of 10 µm can be achieved within a depth of field of 2.4 mm. However, it took about 10 s after imaging to generate a series of images for each eye, limiting its dynamic capabilities. X-ray light field cameras were proposed and simulated by Vigan et al. [93] to reduce the measurement time and dose compared to medical computed tomography (CT) scanners, but are less effective for translucent objects and stay at the simulation level. Combining the monocular depth measurement capability of the light field camera, Shademan et al. [13] applied it to surgical robot positioning and improved the accuracy from 0.5 mm to 0.18 mm compared to the previous use of a 2D camera for planar suturing tasks. In the field of intelligent driving, Ding et al. [94] combined a light field camera with a spectral think and applied it to distance measurement for unmanned vehicles, and the refocused object distances were in general agreement with the actual distances, with an estimated error between 1.5% and 2.7%. The good dynamic performance has also led to the good application of light field cameras in flow diagnostics [95], and the development and application of light field cameras in fluid measurement were described in great detail by S. Shi [22].

Thirdly, light field cameras can not only acquire images, but also obtain three-dimensional information and, therefore, have strong performance in detection and identification. Some typical applications of the light field camera based on this are shown in Figure 16. As mentioned above, Yang et al. [96] used the image information provided by the light field camera with 3D information to achieve the detection of rice plague. Jia et al. [97] used a light field camera for the recognition of 2D dummies such as street signs, where traditional 2D image pedestrian detection cannot effectively distinguish between people and “fake pedestrians”, while the light field camera has the ability to reconstruct in three dimensions and can achieve the recognition of fake pedestrians with a maximum recognition accuracy of 98.33%. However, the need to use commercial image-processing software to cope with the higher complexity of image processing reduces the applicability of the method. Bae et al. [98] used a light field camera for facial-based emotion determination with an average accuracy of 0.85, but the measurement capability of the light field camera was not required.

In conclusion, when compared to traditional vision systems, the light field camera has advantages such as compactness, small size, dynamic measuring capabilities, and so on, and is widely utilized in aerospace, biomedicine, agricultural production, intelligent driving, and other fields. At the moment, the most-significant limitation restricting the applicability of the light field camera is its measurement precision. It has not improved substantially or even declined in comparison to the multi-view system, but in many situations and conditions, the multi-view system has its complexity and inapplicability; thus, the study and application of the light field camera are also very important. First, the light field camera can be designed to maximize the accuracy of the system by designing the appropriate field of view and parameter values. M. Diebold et al. [21] detailed the process of designing the light field setup, including practical issues such as the camera footprint and component size influencing the depth of field and the lateral and range resolution. Second, curved compound eye arrays can be used to overcome the issue of imaging resolution [99]. For example, Woong et al. [100] used curved compound eye technology to improve the resolution of imaging systems, but their measurement capability has yet to be examined. Combining the light field camera approach with the curved compound eye technique may thereby broaden its measurement potential. Finally, research on calibration and reconstruction methods will increase the accuracy of light field camera measurements, allowing them to be used more widely.

## 6. Conclusions

The light field camera meets the needs of small size, speed, and accuracy at the same time and is increasingly used in aerospace, biomedicine, agricultural production, intelligent driving, and other fields. This paper summarized the key technologies of the light field camera in measurement, from calibration methods, reconstruction algorithms, and measurement applications:

1. Calibration methods: This paper provided a comprehensive overview of light field camera calibration approaches, specifically focusing on light ray model construction, light field raw image calibration, and measurement model calibration. The calibration of the light field camera has been extensively studied, covering various calibration parameters, model constructions, and feature-extraction techniques. Calibration parameters such as focal length, aberrations, main lens poses, and microlens array poses have been investigated and calibrated to ensure accurate and reliable light field capture. In the future, the restrictions of pinhole and narrow lens models on the building of light field camera models can be overcome, and other target types, such as screen (the correspondence between screen pixels and camera pixels can be determined by phase calculating [101]), can be used to improve feature-extraction accuracy.

2. Reconstruction algorithms: There are two types of light field reconstruction methods: traditional algorithms and deep-learning-based algorithms. The reconstruction accuracy of traditional algorithms has been substantially improved by extensive study, and it operates well in the presence of occlusion, noise, and so on; however, it suffers from a slow reconstruction speed. The deep learning algorithm tackles the problem of the traditional approach’s lower efficiency, but has somewhat worse stability and accuracy. In the future, it is critical to enhance the algorithm’s efficiency in terms of hardware and software while maintaining accuracy. With more studies on approaches such as neural radiance fields (NeRFs) [102,103], which reconstructs 3D images from multi-view images, it can be integrated with light field 3D reconstruction to boost algorithm performance even more.

3. Measurement applications: The light field camera has unique measurement advantages and is widely used in the measurement field. This paper classified and summarized the measurement applications of light field cameras based on their characteristics. In the future, with further improvements in measurement accuracy, light field cameras will achieve even wider applications. They will play an important role in applications such as internal wall measurement, endoscopic measurement, and underwater environmental inspection, where special requirements are placed on the measurement system.

## Figures and Tables

**Figure 1 sensors-23-06812-f001:**
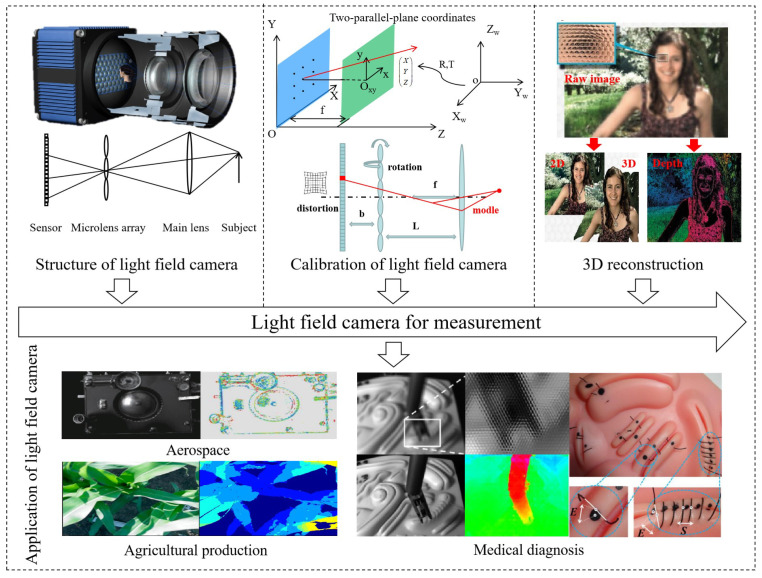
Key technologies of light field camera in measurement [10,11,12,13].

**Figure 2 sensors-23-06812-f002:**
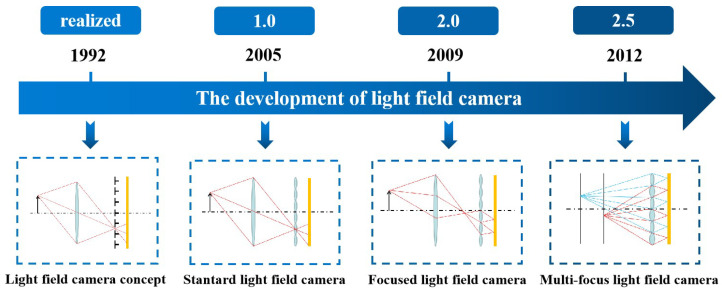
The development of the light field camera.

**Figure 3 sensors-23-06812-f003:**
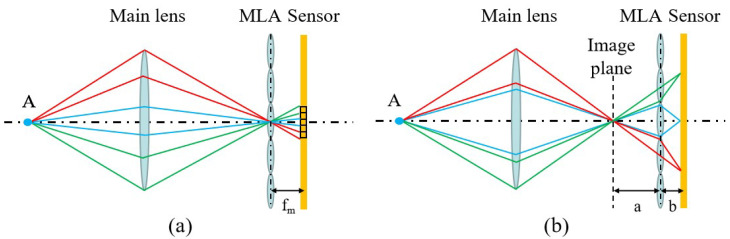
Measurement fundamentals of the light field camera. (**a**) The fundamental of the standard light field camera. (**b**) The fundamental of the focused light field camera.

**Figure 4 sensors-23-06812-f004:**
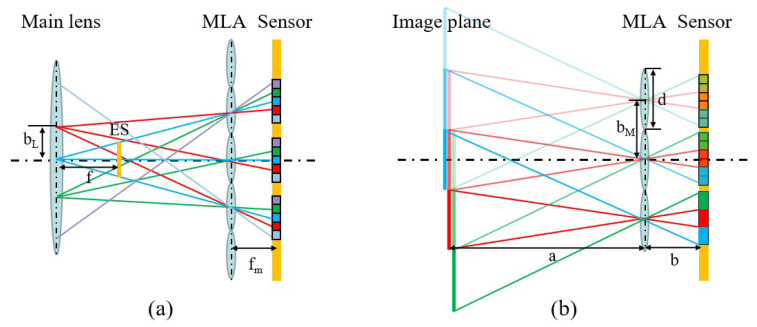
The schematic diagram of the equivalent camera array. (**a**) The diagram of the standard light field camera. (**b**) The diagram of the focused light field camera.

**Figure 5 sensors-23-06812-f005:**
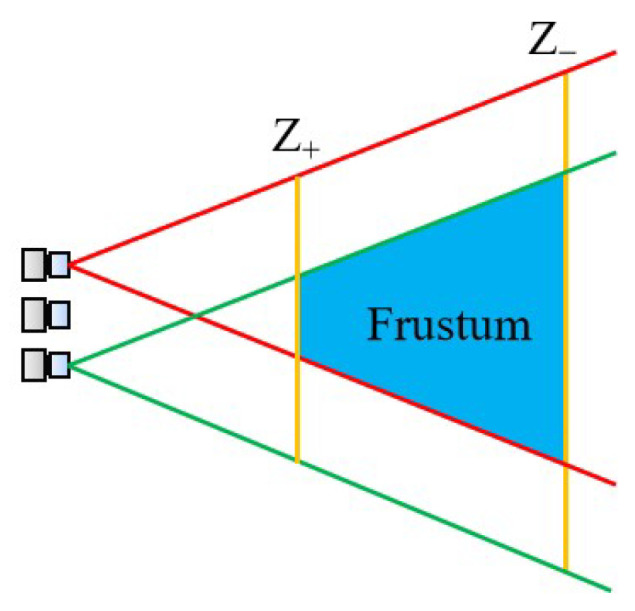
The frustum resulting from a designed light field camera setup [21].

**Figure 6 sensors-23-06812-f006:**
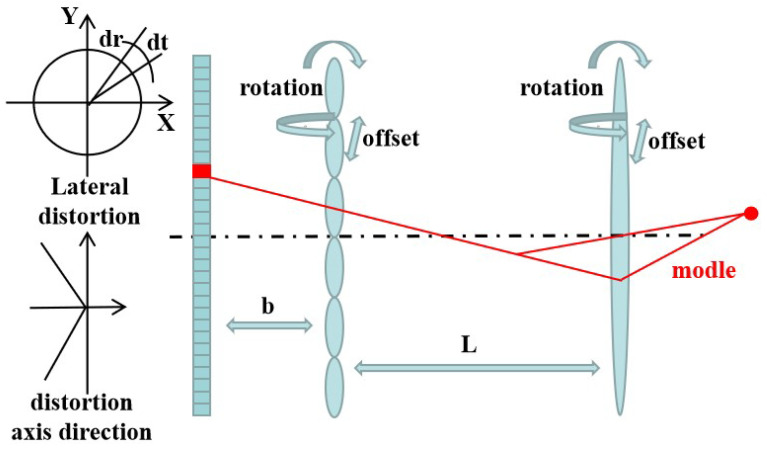
The light model of the light field camera. (L: distance from main lens to microlens array, b: distance from microlens array to image sensor, dr: radial distortion, dt: tangential Distortion.).

**Figure 7 sensors-23-06812-f007:**
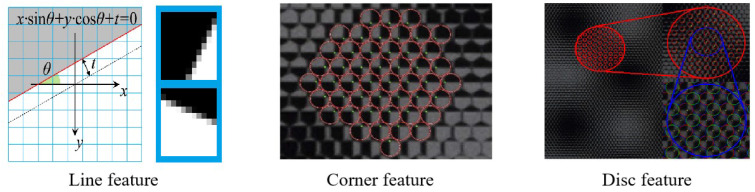
Feature extraction from raw images [31,33,34].

**Figure 8 sensors-23-06812-f008:**
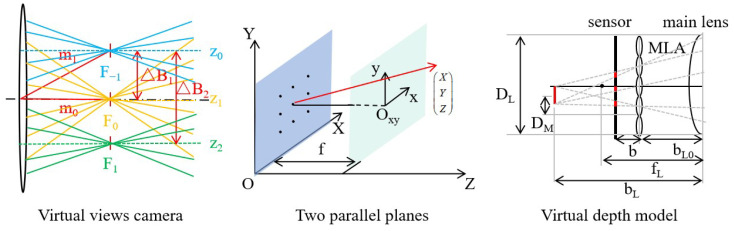
Model construction of light field camera [35,36,39].

**Figure 9 sensors-23-06812-f009:**
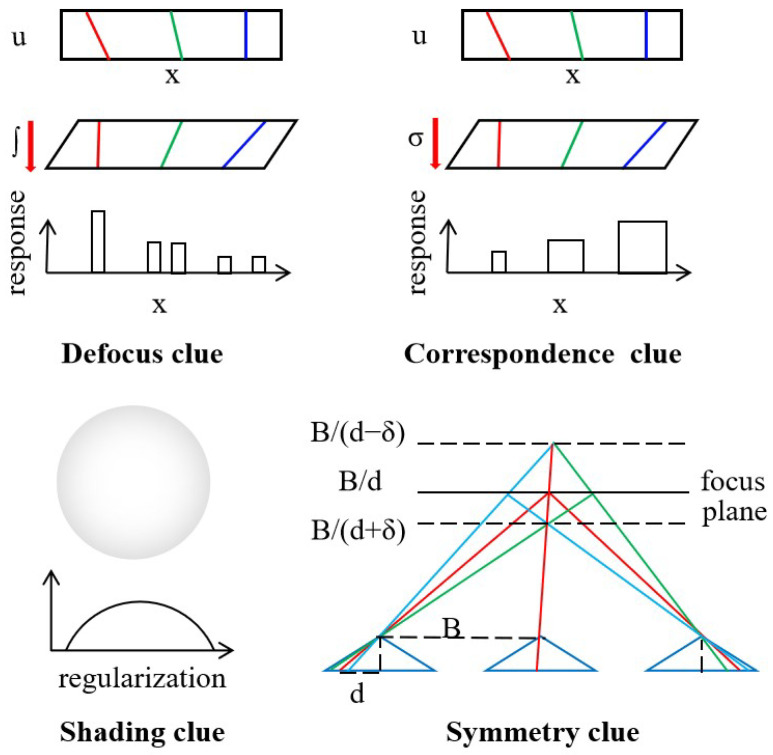
Clues in light field camera reconstruction [59,60,61]. (*d* and δ: the focal slice shifted from the true disparity *d* by δ, *B*: camera baseline length.)

**Figure 10 sensors-23-06812-f010:**
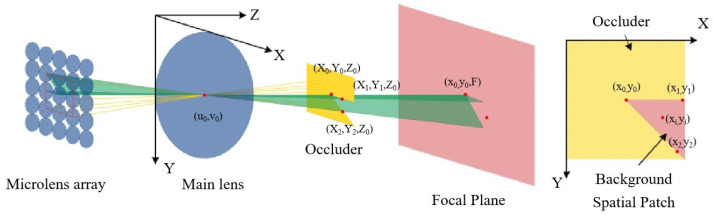
The light field camera model with occlusion [64].

**Figure 11 sensors-23-06812-f011:**
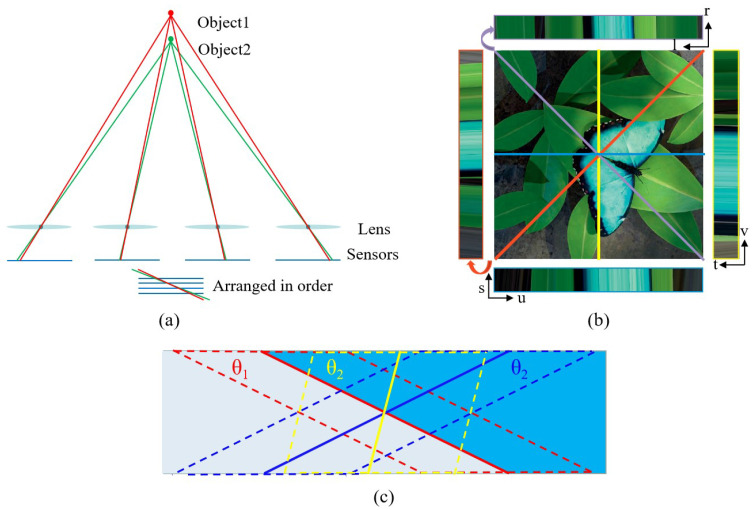
Measurement principle and application of epipolar-based reconstruction. (**a**) The principle of epipolar-plane-based method. (**b**) Multi-orientation epipolar plane images [70]. (**c**) The schematic diagram of the SPO operator [69]. (θ: different angles of parallelogram).

**Figure 12 sensors-23-06812-f012:**
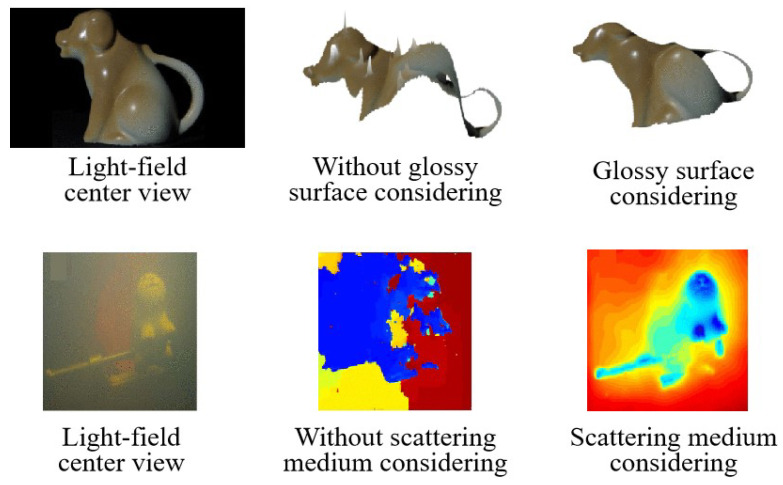
Results for glossy-surface- and underwater-scattering-reconstruction methods [74,75].

**Figure 13 sensors-23-06812-f013:**
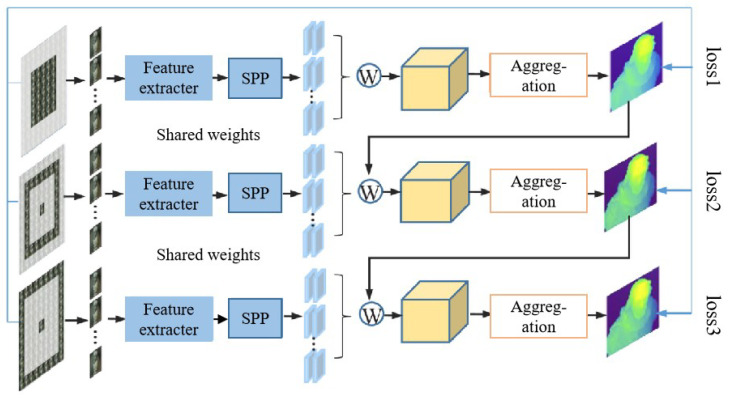
The overall architecture of cascaded-cost-volume network [82].

**Figure 14 sensors-23-06812-f014:**
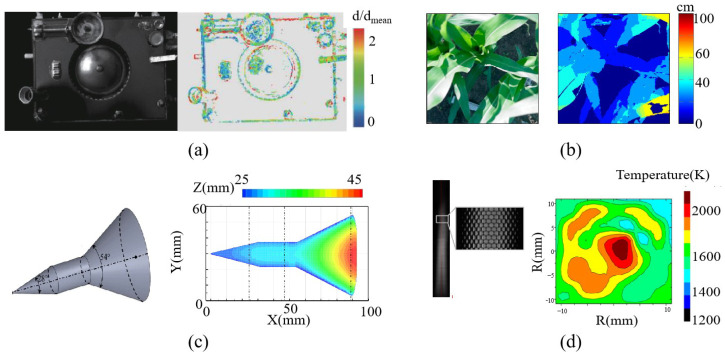
Measurement applications based on cost and volume control, as well as convenience and stability. (**a**) Measurement for non-cooperative spacecraft [10]. (**b**) Measurement for external structure of plants in the field [12]. (**c**) Measurement for fully aerodynamic models in large wind tunnel experiments [90]. (**d**) Measurement for flame temperature [88].

**Figure 15 sensors-23-06812-f015:**
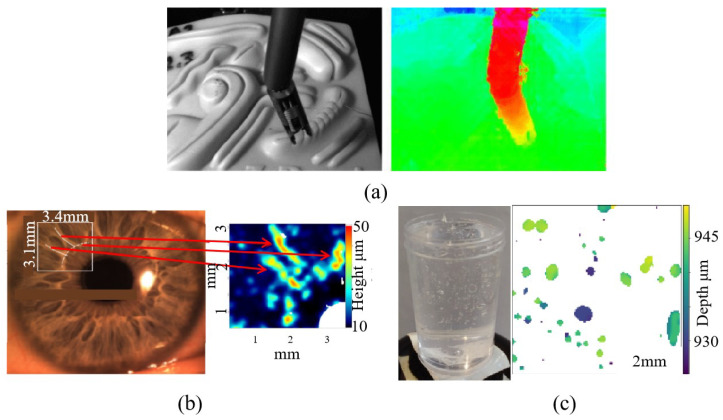
Measurement applications based on dynamic capabilities. (**a**) Measurement for surgical robot positioning [13]. (**b**) Measurement for human iris [92]. (**c**) Measurement for air bubbles in gel through X-ray light field camera [93].

**Figure 16 sensors-23-06812-f016:**
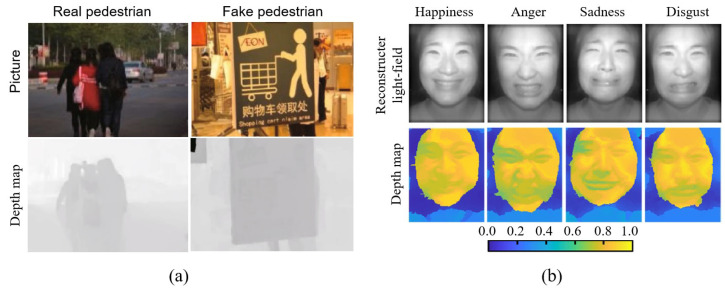
Measurement applications based on detection and identification capabilities. (**a**) Distinguishing between fake pedestrians and real pedestrians [97]. (**b**) Determination of emotion based on facial measurements [98].

**Table 1 sensors-23-06812-t001:** Comparison of the advantages and disadvantages of measurement systems.

System	Advantages	Disadvantages
Single camera	Simple system, small size, and light weight	Poor depth recovery
Multi-view system	High measurement accuracy and retention of features such as color and texture	Cumbersome calibration and large system size
LiDAR	Long measuring distance, high measuring accuracy, and low influence by light	High cost and low sampling accuracy
Light field camera	Small size, light weight, good dynamic performance, and dense view	Low measurement accuracy and high data volume

**Table 2 sensors-23-06812-t002:** Comparison of the parameters of standard and focused light field cameras [18].

Type of Light Field Camera	Angular Resolution (Depth Resolution)	Spatial Resolution (Lateral Resolution)	Depth of Field	Reconstruction Accuracy
Standard	high	low	high	low
Focused	low	high	low	high

**Table 3 sensors-23-06812-t003:** Comparison of reconstruction algorithms on datasets (Badpix0.07/MSE) [82].

	Wang	Williem	Shin	Tsai	Peng	Liu
Buddha	11.68/1.94	3.21/0.64	1.55/0.36	2.02/0.33	11.55/1.14	4.54/0.33
Papillon	29.97/0.83	7.33/0.65	35.56/6.12	34.96/5.07	30.30/5.32	27.10/1.06
Stilllife	59.45/84.61	14.4/1.26	11.37/2.43	11.78/14.01	42.05/17.28	8.97/5.52
Avg	33.7/29.13	8.31/0.85	16.16/2.97	16.25/6.47	27.97/7.91	13.54/2.30

**Table 4 sensors-23-06812-t004:** Comparison of traditional algorithms and deep learning algorithms.

Algorithms	Precision	Efficiency	Stability	Dataset
Traditional	high	low	more stable	No
Deep learning	low	high	less stable	Yes

**Table 5 sensors-23-06812-t005:** Light field camera product parameters [11,85].

Company	Camera Model	Field of View X (mm )	Field of View Y (mm)	Depth of Field Z (mm)	Lateral Resolution (μm)
Raytrix	R26	500	500	500	250
	R12micro	0.22	0.15	0.01	0.25
VOMMA	V15-L	1532	854	1850	500
	VA6-H	1.5	1.1	0.042	2.2

## Data Availability

Not applicable.

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
