# Peer review of "Measurement Technologies of Light Field Camera: An Overview"

_sensors, 2023, doi:10.3390/s23156812_

Round 1

Reviewer 1 Report

Review of manuscript sensors-2460833-peer-review-v1

Title: Measurement Technologies of Light Field Camera: An Overview

In this work, a comprehensive description of the basic principles, main calibration and rendering techniques and measurement applications of the Light Field camera is presented. In addition, the authors point out future research opportunities in this field, considering the advantages and limitations of the existing approaches. The strong aspects of this papers relate to the carefully developed and well-structured analysis of the available calibration and reconstruction techniques. In the related sections, the rationale is developed in an organized way and the ideas are well connected. On the other hand, the section describing the imaging principle of the light field camera is not giving a very clear picture of the underlying physical principles, specially to those who don’t have a deep knowledge in this area. It should be noted that, since the focus lies on measurement technologies, the potential reader may work in a related area, for example, LiDAR, digital holography, multi-view systems, etc, and have no specific understanding of measurement principles of the Light Field systems. Thus, a clearer explanation of the principles governing the differences between the different types of Light Field camera (Standard, Focused, and Multifocus) would be desirable.

Some minor corrections:

Page 5, line 140: “vrey” should be replaced by “very”

Page 5, line 148: Fmla represents the aperture value -> this parameter is commonly named “f-number” of an optical system and is inversely proportional to the numerical aperture. So, “aperture value” is not a correct designation.

Page 5, lines 146-152: It would be advisable to review this (and the following) part. It is very hard to follow due to some loose terminology.

Page 7, lines 163-170: Again, some designations can be misleading. Spatial resolution, Rs, is the name given in this section to a non-dimensional parameter that is a scaled version of the number of pixels based on geometrical optics considerations. It can be mistaken for lateral resolution in length units, if we consider that Rd represents the depth resolution and has units of length.

Page 9, line 251: First mention of the acronym EPI without definition (a definition of EPIs is given in page 12, line 357).

Section 5: Measurement applications.

Some remarks on this section: Given the title of this paper, one would expect a deeper description of some specific measurement techniques and how these relate to the aforementioned tradeoffs of resolution and field of view, or computational rendering complexity.

Author Response

We would like to express our sincere thanks for your constructive and positive comments. We have carefully reviewed them and tried our best to improve the manuscript according to your comments point by point. Please see the attachment for specific changes.

Reviewer 2 Report

The authors present a review of the state-of-the-art related to emergent light field cameras.

Concepts are clearly explained and application scenarios and processing algorithms are presented.

I do miss a table benchmarking relevant contributions in the field of other authors and/or manufacturers. The table should compare the specifications of the different sensors and discuss their capabilities.

The authors should define clearly the parameters to benchmark different light field cameras.

The paper is focused on the description of the optical system. However, I miss details about how the image sensor must be selected to implement a light field camera. What are the optimum pixel pitch, number of pixels, frame rate, etc?

There are some typos in the text. For instance, in line 111 'chapter 4' is referred to the reader. It seems that this text was copied from a previous publication. Please, correct them.

There are some typos in the text. For instance, in line 111 'chapter 4' is referred to the reader. It seems that this text was copied from a previous publication. Please, correct them.

Author Response

(The authors gave the same response as above.)

Author Response

(The authors gave the same response as above.)

Round 2

Reviewer 2 Report

The authors have addressed properly all my comments.

English is acceptable.